# Effects of Façades on Urban Acoustic Environment and Soundscape: A Systematic Review

Alvaro Balderrama [1,2,*], Jian Kang [3], Alejandro Prieto [4], Alessandra Luna-Navarro [1], Daniel Arztmann [2] and Ulrich Knaack [1]

1   Architectural Façades and Products Research Group, Department of Architectural Engineering and Technology, Faculty of Architecture and the Built Environment, Delft University of Technology, Julianalaan 134, 2628 BL Delft, The Netherlands
2   IDS Institute for Design Strategies, Detmold School of Architecture and Interior Architecture, Technische Hochschule Ostwestfalen-Lippe, Emilienstraße 45, 32756 Detmold, Germany
3   UCL Institute for Environmental Design and Engineering, The Bartlett, University College London (UCL), Central House, 14 Upper Woburn Place, London WC1H 0NN, UK
4   Facultad de Arquitectura, Arte y Diseño, Escuela de Arquitectura, Universidad Diego Portales, Av. República 180, Santiago 8370074, Chile
*   Correspondence: a.balderrama@tudelft.nl

**Abstract:** Façades cover a significant amount of surfaces in cities and are in constant interaction with the acoustic environment. Noise pollution is one of the most concerning burdens for public health and wellbeing; however, façade acoustic performance is generally not considered in outdoor spaces, in contrast to indoor spaces. This study presents a systematic literature review examining 40 peer-reviewed papers regarding the effects of façades on the urban acoustic environment and the soundscape. Façades affect sound pressure levels and reverberation time in urban spaces and can affect people's perception of the acoustic environment. The effects are classified into three groups: Effects of façades on the urban acoustic environment, including sound-reflecting, sound-absorbing and sound-producing effects; Effects of façades on the urban soundscape, including auditory and non-auditory effects; Effects of the context on the acoustic environment around façades, including boundary effects and atmospheric effects.

**Keywords:** façade; building envelope; acoustics; acoustic environment; soundscape; urban comfort

## 1. Introduction

Since the middle of the twentieth century, the world population has more than tripled in size, rising from around 2.5 billion in 1950 to almost 7.9 billion in 2021 [1], leading to an unprecedented expansion and densification of cities. Populated urban environments are generally more affected by noise pollution as indicated by noise maps developed in accordance with the Environmental Noise Directive (2002/49/EC) [2], which require EU member states to make a yearly calculation of the exposure to environmental noise in cities and to provide action plans. As indicated by [3], urban air, water, and noise pollution can have substantial effects on the mental health of urban populations. For example, living close to major streets or airports increases exposure to traffic noise pollution and is associated with higher levels of stress, aggression, and an increased risk of impaired mental health. According to European Environment Agency [4] an estimated 113 million people are affected by long-term day-evening-night traffic noise levels of at least 55 decibels. In most European countries, more than 50% of inhabitants within urban areas are exposed to road noise levels of 55 dB or higher during the day-evening-night period.

The acoustic environment (also called the sound environment, aural environment, or sonic environment among other terms), can be understood as the "sound at the receiver from all sound sources as modified by the environment" [5]; it can be indoor or outdoor, as

well as measured, simulated, experienced, or remembered. Since the creation of the decibel unit in the 1920s, measurements of the sound pressure level have been continuously used to assess the acoustic environment and noise (unwanted sound) in an objective manner.

Aside from physical sound measurements, people's perception of the acoustic environment and its psychological and physiological implications are crucial to soundscape research, which dates back to the work of R. Murray Schafer [6,7] and his contemporaries in the 1960s and 1970s [8,9]. Since then, the interest on the soundscape has developed into an interdisciplinary research field of increasing interest. In 2014, the International Organization of Standardization published the first standard on soundscape, ISO 12913-1:2014 [5], which defined the term as "the acoustic environment as perceived or experienced and/or understood by a person or people, in context", published along with a series of definitions and a conceptual framework, followed by methods for data collection and analysis [10,11].

As cities grow and densify, there are generally more people and more sound sources, such as vehicles, air traffic, construction machinery, and leisure activities, among others, surrounded by buildings and other infrastructure. Façades, generally covering the vertical surfaces of building envelopes, are constantly exposed to urban sounds playing a role in the composition of the urban soundscape. However, as described by [12], the impact of urbanization and the influence of the façades on the urban soundscape have not yet been implemented into architectural façade design. The acoustic performance of façades becomes even more important in noisy surroundings due to the measurable negative effects on people's health caused by the increasing average noise levels. A systematic literature review from 2020 [13] studied building envelope design strategies for better urban acoustic environments, considering façades, roofs, balconies, and ceilings, among others. The study identifies a series of design strategies from empirical research in individual studies and shows that a standardized assessment of façade acoustic performance outdoors is not yet defined.

The main purpose of this systematic review is to explore existing published research related to façade design, the acoustic environment and the soundscape in urban environments in order to provide an overview of the state-of-the-art and answer the following research questions: (1) How is the study design of experiments that focus on façade-related effects on the acoustic environment and the soundscape? (2) What parameters of façades and their contexts are involved in the production of effects on the acoustic environment and the soundscape? (3) What are the effects of façades on the acoustic environment and the soundscape? This paper gathers data from articles that report how building façades have affected the acoustic environment and/or the soundscape (people's perception of the acoustic environment). Section 2 describes the process of searching, identifying, and selecting the literature to be examined. Section 3 shows the result of the search and the synthesis of data extracted in three tables regarding study design, reported effects, façade parameters and contextual parameters. Section 4 discusses the classification of effects. Section 5 presents the conclusions. Section 6 includes the limitations of the study.

## 2. Methods

The literature studied in this review was collected by considering the applicable procedures of the preferred reporting items for systematic reviews and meta-analyses [14], also known as PRISMA. No pre-defined protocol has been registered. To the best of the authors' knowledge, there are no previous systematic reviews focused on the effects of façades on the urban acoustic environment and the soundscape.

### 2.1. Search Strategy and Eligibility Criteria

The literature review searched peer-reviewed papers published in English related to effects produced by building façades on the urban acoustic environment, and if applicable, on the soundscape. The identification of studies was carried out in October 2021 and surveyed the bibliographic databases of Scopus, Science Direct, and Web of Science. The search terms designated intend to intersect the fields of façade design ("façade" or "building

envelope"), outdoor urban environments ("urban" or "city" or "outdoor"), acoustics and soundscape ("acoustics" or "sound" or "soundscape" or "noise"). To be eligible for the systematic review, the literature sources needed to report physical changes to the acoustic environments and/or changes in people's perception of the acoustic environment due to façade properties.

*2.2. Data Extraction*

A data extraction sheet was created to organize information from each of the 40 studies regarding their study design, façade parameters, and contextual parameters. Since the reviewed literature was gathered from different fields (environmental acoustics, building acoustics, façade acoustics, and soundscape research), the heterogeneous results were not quantified through meta-analysis, but rather through qualitative synthesis to represent the search results and answer the research questions.

**3. Results**

The search in three databases returned 833 results, from which 225 records were duplicates, and therefore removed. The remaining 608 abstracts were screened and 531 records were excluded for not relating to façades or acoustics, and/or for not taking place outdoors. The full text of the remaining 77 papers was assessed and 37 records were excluded due to the eligibility criteria. The final number of papers included in this review is 40 (forty) as shown in Figure 1.

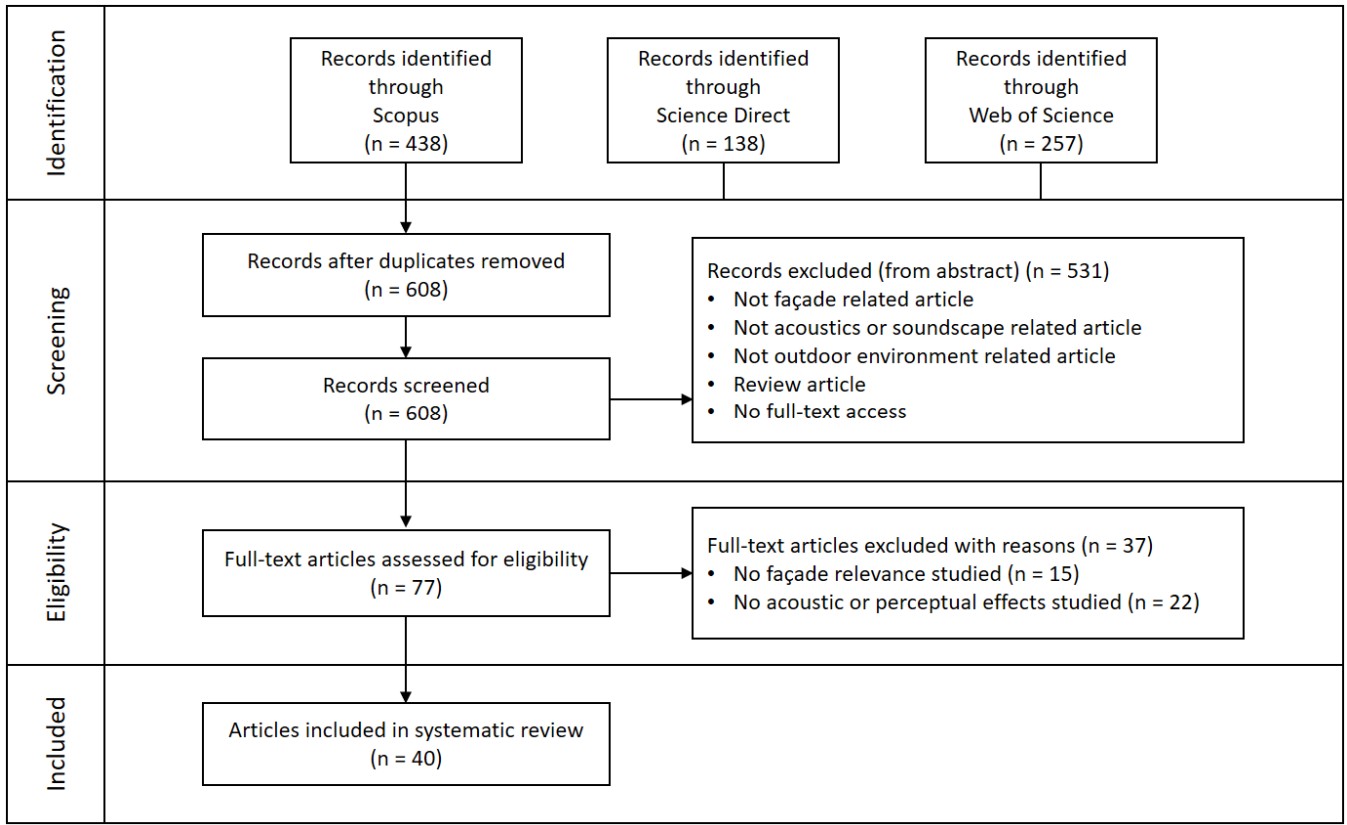

**Figure 1.** Flow diagram of the selection process and the final number of articles included in the systematic review (n = 40).

Figure 2 presents the number of publications published per year from 1994 to 2021 (n = 40) differentiating between studies that focus on noise abatement in which the decibel level is the main descriptor (n = 37) and studies that focus on the soundscape in which people's reported perception of the acoustic environment is the main descriptor (n = 3).

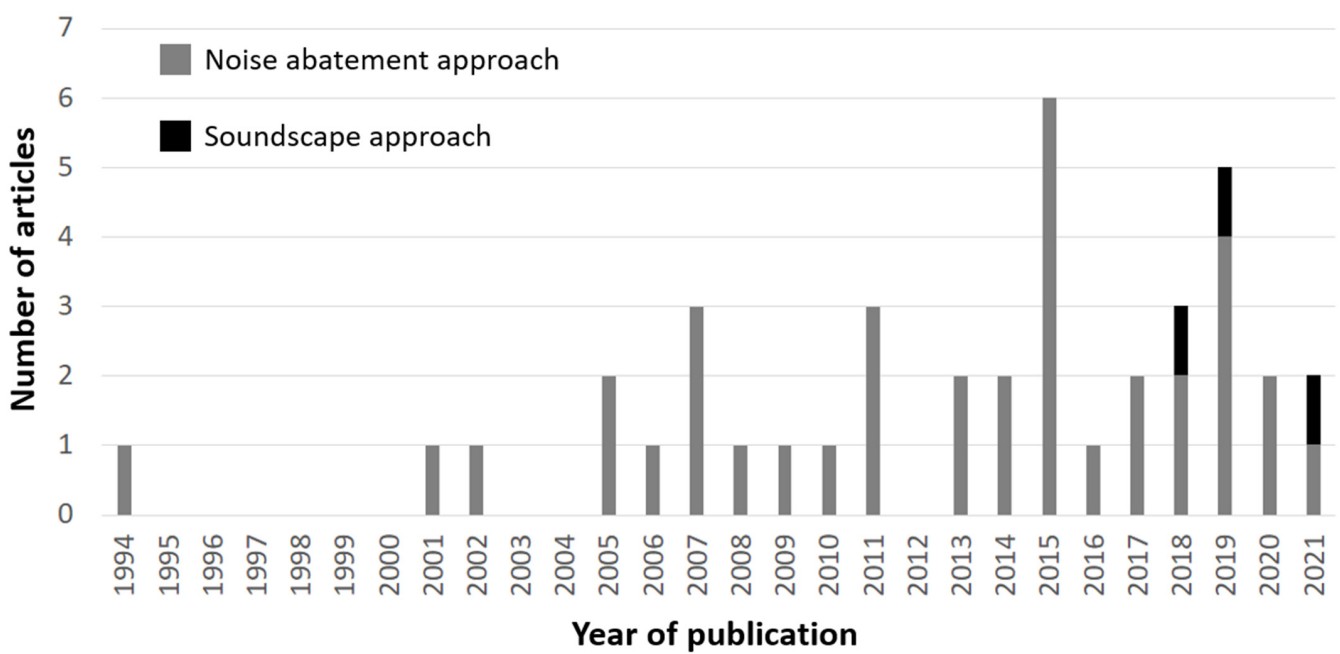

**Figure 2.** Amount of publications per year included in the systematic review.

The data extracted from 40 studies [15–54] are presented in reverse chronological order of publication, from 2021 to 1994, in two tables. In Table 1, following the first column of references is "Study design", considering the location, general method used, objective measures used, and perceptual attributes considered (if any). Additionally, "Reported effects" present a compilation of excerpts about effects related to façades. Table 2 presents extracted data regarding "Façade parameters", such as the geometrical features of height and depth of the façade being studied as well as the façade materials, and "Contextual parameters", which considers three aspects: sound, path, and receiver. "Sound" includes the sound source, frequency content, and if background noise is considered. "Path" includes the physical boundary conditions of the urban context (morphology and materials), and atmospheric conditions. "Receiver" includes the type (e.g., a person, a measurement instrument, or a simulation), and its position in relation to the façade. In cases in which specific data were not clearly identified, data extraction was not applied, "n/a". The PRISMA Checklist supporting this study can be accessed in Supplementary Materials.

**Table 1.** Summary of study design and reported effects in the reviewed studies (n = 40) in reverse chronological order of publication. Abbreviations: not-applicable (n/a); field measurement (FM); scale model measurement (SM); laboratory experiment (LE); simulation (SI); number of participants (n).

| Reference | Study Design | | | | Reported Effects |
|---|---|---|---|---|---|
| | Location | Method | Objective Measure | Perceptual Attribute | |
| Masullo et al., 2021 [15] | (a) Naples, Italy; (b) Barcelona, and (c) Valencia, Spain | FM, LE (n = 30) | Leq | Loudness, Noise Annoyance, Visual Pleasantness | • Visual elements that are more integrated (aesthetically) on the façade of historical buildings can reduce the auditory and visual impact of these elements.<br>• Well-integrated and pleasant elements led to a lower auditory perception of the loudness and noise annoyance than less integrated elements. |
| Niesten et al., 2021 [16] | Delft, The Netherlands | SI, FM, SM (1:50) | SPL, Leq | n/a | • Sound reflective design (inclined upwards) reduced SPL by up to 9.3 dB(A).<br>• Sound-absorbing design reduced SPL inside the courtyard area by up to 5.5 dB(A).<br>• Combined design (geometry and absorption) reduced SPL by up to 6.7 dB(A). |
| Montes González et al., 2020 [17] | Don Benito, Spain | SI, FM | SPL, Leq | n/a | • Cars parked near façades act as shielding between the sound source and façade. Differences up to 4 dB(A) in sound levels were found between situations without and with cars parked, in some cases, up to 8 dB(A). |
| Cabrera et al., 2020 [18] | (a) Berkeley, USA; (b) Sydney, Australia; (c) Hong Kong | FM, SI | SPL | n/a | • Acoustic retroreflection (when sound is reflected back to the source) occurs due to façade geometry and is most prominently in the high-frequency range.<br>• While retroreflections are measurable, they are not necessarily audible in noisy environments. |
| Hupeng et al., 2019 [19] | Harbin, China | SI | Sound attenuation, RT30, EDT | n/a | • Sound attenuation is linearly correlated with sound propagation distance.<br>• The mean reverberation time increases with the increasing mean façade height, sidewalk width, and cross-sectional enclosure. |
| Leistner et al., 2019 [20] | n/a | SI | Insertion loss | n/a | • Sound level reduction due to façade absorption. Insertion loss is from 58.6 dB to 52.7 dB. |
| Yu et al., 2019 [21] | Tianjin, China | SI, FM | SPL, RT | n/a | • Scattering coefficient from building façades affects the sound pressure level and reverberation time in street canyons. |
| Taghipour et al., 2019 [22] | Dübendorf, Switzerland | LE (n = 27) | SPL | Short-term acoustic comfort | • Effects of several variables (façade absorption, type of sound, observer position) on short-term acoustic comfort<br>• Façade absorption was found, generally, to increase acoustic comfort. Too much absorption, however, was not found to be helpful.<br>• Significant differences observed between acoustic comfort at distinct observer/listener positions. |
| Badino et al., 2019 [23] | Turin, Italy | SI | SPL | n/a | • Geometrically optimized façade cladded with sound absorbing materials can decrease noise level by up to 10 dB over the façade and up to 3 dB over the opposite one.<br>• Up to 1 dB decrease in the mean level over the entire façade achieved with balconies that have a depth of 1.5 m compared to 0.9 m, with a maximum abatement of 2.8 dB at the highest floor. |
| Calleri et al., 2018 [24] | Turin, Italy | SI, FM, LE (n = 31) | SPL, RT30, C50, DRR | Space wideness | • Absorption and scattering coefficients of façade upholsteries and listener position have an influence on the acoustic characteristics of the square and the subjective assessment of its wideness through auditory perception.<br>• T30 was the most influential parameter on perceived space wideness. |
| Jones and Goehring, 2018 [25] | Pacific Northwest, USA | SI | SPL | n/a | • Pressure fluctuations that result from wind produce noise on the façade's perforated panels.<br>• Frequency and audibility are influenced by hole diameter, hole spacing, panel thickness, wind velocity, turbulence intensity, and wind angle of incidence.<br>• The corners and the top of the building are the most prone to wind-induced noise. |
| Montes González et al., 2018 [26] | n/a | FM, SI | SPL | n/a | • Noise maps neglect the shielding effects of cars parked in front of buildings. Differences of up to 3 dB with and without parked cars in specific heights along the façade. |
| Qu and Kang, 2017 [27] | North Oxford; Rotherham; Greater London, UK | SI | Leq | n/a | • Considering noise from wind turbines, built environment morphology creates large variations in sound levels (up to 10 dB(A)) around dwellings at building scale in the distant range of 400–1000 m from the source. |

**Table 1.** *Cont.*

| Reference | Study Design | | | | Reported Effects |
|---|---|---|---|---|---|
| | **Location** | **Method** | **Objective Measure** | **Perceptual Attribute** | |
| Flores et al., 2017 [28] | (a) Madrid, Spain; (b) Pisa, Italy | FM | Leq | n/a | • Considering aircraft noise, the orientation of buildings toward fight paths influences sound pressure levels on façades. |
| Echeverria Sanchez et al., 2016 [29] | Ghent, Belgium | SI, FM | SPL | n/a | • Building shape can be responsible for variations of up to 7.0 dB(A) on the pedestrian level.<br>• Building façade design can reduce the average exposure at windows with 12.9 dB(A).<br>• Street geometry can enhance the positive effect of low barriers to 11.3 dB(A) along sidewalks.<br>• Building geometry mainly influences noise levels along the façades, whereas geometrical changes to noise barriers next to the source have a higher relevance for pedestrians and at the windows of lower floors. |
| Jang et al., 2015 [30] | n/a | SI, SM (1:10) | SPL | n/a | • Noise reduction due to the vegetated façades was less than 2 dB at the pedestrian level in a two-lane street canyon.<br>• The noise reduction effect due to the absorption performance was more effective in low-frequency bands than in high-frequency bands, on the basis of 1 kHz. |
| Can et al., 2015 [31] | n/a | SI | SPL | n/a | • Overall sound level increase due to diffusion by up to 10 dB, according to the street geometry and acoustical parameters.<br>• Diffusion by façades and fittings impact sound attenuation within street canyons. |
| Guillaume et al., 2015 [32] | Nantes, France | SI, FM | SPL, EDT | n/a | • Beneficial effect of greening building façades and rooftops in terms of both acoustic level and sound decay time indicators at low-frequency third-octave bands.<br>• The effect of vegetation on sound levels at the façade presents a 5 dB reduction. |
| Sakamoto and Aoki, 2015 [33] | Japan | SI, SM (1:20) | Insertion loss | n/a | • Flat eaves in a horizontal direction attached on upper and lower positions of the story have both positive and negative effects on noise propagation from a source to a receiver: a shading effect by the lower eave and a reflection effect by the upper eave. As a result, the noise reduction is not so high.<br>• Making an upper eave inclined is effective countermeasure for noise reduction because the inclined eave reflects the incident sound outward from the surrounding surface of the building. The noise reduction effect is higher as the receiving point is higher.<br>• Louvers with horizontal short fins are also effective at high stories. |
| Jang et al., 2015 [34] | n/a | SM (1:10) | Insertion loss, RT | n/a | • Vegetated façades mitigated the overall noise level up to 1.6 dB(A) in the street canyon, and greening façades were effective to reduce low frequency noise levels below 1 kHz.<br>• Vegetated façades in street canyon effectively reduced noise below 630 Hz in courtyards. |
| Hao and Kang, 2014 [35] | Assen, the Netherlands | SI | L10, L50, L90, Lavg | n/a | • Within the 1000 m horizontal distance of flight path to a site, urban morphology plays an important role in sound propagation, especially for the buildings with high sound absorption façades, where the variance of average noise level attenuation among different sites is about 4.6 dB at 3150 Hz.<br>• The effect of a flight altitude of 200–400 ft (60–120 m) on average noise level attenuation is about 2 dB at both 630 Hz and 1600 Hz in open areas. |
| Silva et al., 2014 [36] | Braga, Portugal | SI | Leq | n/a | • Influence of the urban form on the noise exposure of building façades. |
| Van Renterghem et al., 2013 [37] | n/a | SI | Insertion Loss | n/a | • Fully vegetating the source canyon does not give additional benefits compared to only treating the upper half in case of soft bricks, while additionally 1 dB(A) can be gained in case of rigid bricks. The presence of wall vegetation in the lower part only results in a rather limited insertion loss.<br>• Insertion loss of 4.4 dB(A) in case of fully vegetated source canyon façades.<br>• Fully vegetating the receiving canyon has an additional advantage compared to only treating the upper half. |
| Thomas et al., 2013 [38] | Ghent, Belgium | FM, SI | SPL, RT | n/a | • An increase in average height generally leads to an increase in SPL. For positions further from the source, the change in the SPL over the average height is bigger than for positions close to the source. |

**Table 1.** *Cont.*

| Reference | Study Design | | | | Reported Effects |
|---|---|---|---|---|---|
| | Location | Method | Objective Measure | Perceptual Attribute | |
| Hornikx and Forssén, 2011 [39] | n/a | SI | SPL | n/a | ● Noise can increase up to 10 dB(A) in an open courtyard compared to in a closed courtyard. To counteract the impairment due to the façade opening, absorption can be applied to the opening walls at the same time as additional façade absorption.<br>● Opening on façades causes the average level to be up to 7.2 dB(A) higher for the façade receiver positions and up to 10.1 dB(A) higher for the courtyard receiver positions.<br>● Effect of façade absorption of up to 1.5 dB(A). |
| Tang and Piippo, 2011 [40] | n/a | SM (1:4) | SPL, RT | n/a | ● The increase in sound level due to the presence of an opposite vertical wall can be as high as 8 dB, probably because of the multiple images effect and the increased reverberation.<br>● As the inclination of one of the wall decreases, the reverberation strength decreases quickly and the sound field becomes less uniform. |
| Oliveira and Silva, 2011 [41] | n/a | SI | Leq | n/a | ● The average values of Leq will increase as the number of floors increases. |
| Okada et al., 2010 [42] | n/a | SI, SM (1:40) | SPL | n/a | ● Sound pressure levels increase with increasing height of buildings.<br>● Sound pressure levels increase with the viaduct road width and building density. |
| Hornikx and Forssén, 2009 [43] | Göteborg, Sweden | SI | SPL | n/a | ● A change in the façade absorption coefficient leads to a reduction of around 4 dB(A) for most canyon observer positions.<br>● Façade absorption is the most effective when placed in the upper part of the canyon. |
| Hornikx and Forssén, 2008 [44] | n/a | SM (1:40) | RT10, SPL | n/a | ● Sound propagation in parallel canyons affected by inserting absorption and diffusion patches in the façades of the source canyon. The level differences between rigid façades and applied absorption or diffusion patches are larger in the shielded canyon than in the directly exposed street canyon. |
| Onaga and Rindel, 2007 [45] | n/a | SI | SPL, RT | n/a | ● The effect of façade scattering on the SPL appears to increase at short distances and decrease at great distances. The range of the increase in SPL is larger in high-façade streets. In low-façade streets, the primary effect of scattering on SPL is a decrease in SPL.<br>● In low-façade streets, the reverberation time is determined by the sum of absorption coefficient and scattering coefficient. In contrast, in high-façade streets, the reverberation time is determined by the absorption coefficient. |
| Heimann, 2007 [46] | n/a | SI | SPL | n/a | ● In parallel streets, façades of flat-roof buildings are quieter than those of hip-roof buildings despite equal cross-cut areas. The wind effect (resulting in quieter upwind and louder downwind façades) is more pronounced for hip-roof buildings. In the case of parallel streets, upwind façades are slightly louder than downwind façades because they are simultaneously exposed to downwind propagating sound from the next parallel street. |
| Lee et al., 2007 [47] | Seoul, Korea | SI, FM, SM (1:50) | SPL, RT20 | n/a | ● Studying façades with balconies on a scale model, the combination of absorbing surfaces on the parapet and inclined ceiling provided a maximum noise reduction of 16 dB at 1 kHz. |
| Van Renterghem et al., 2006 [48] | n/a | SI | SPL | n/a | ● Diffusely reflecting façades and balconies lead to an important increase in shielding compared to flat façades. Near 1000 Hz, about 10 dB in shielding is gained for the profiled façade (introducing recesses by windows and protrusions by windowsills, together with a roughened wall)<br>● In case of downwind sound propagation, shielding decreases by an important degree compared to a non-moving atmosphere. With increasing incident wind speed and with increasing frequency, shielding decreases.<br>● In case of upwind sound propagation, turbulent scattering plays an important role and shielding does not increase compared to a non-moving atmosphere. |
| Thorsson and Ogren, 2005 [49] | Stockholm, Sweden | SI | Leq, SPL | n/a | ● Absorption onto building façades will give lower levels at shielded positions.<br>● Absorptive material can reduce the noise levels by at least 5 dB when located inside the canyons. |

**Table 1.** *Cont.*

| Reference | Study Design | | | | Reported Effects |
|---|---|---|---|---|---|
| | Location | Method | Objective Measure | Perceptual Attribute | |
| Kang, 2005 [50] | n/a | SI | SPL, RT, EDT | n/a | • The SPL in far field is typically 6–9 dB lower if the (urban) square side is doubled; 8 dB lower if the height of building façades is decreased from 50 m to 6 m (diffuse boundaries); 5 dB (diffuse boundaries) or 2 dB (geometrical boundaries) lower if the length/width ratio is increased from 1 to 4; and 10–12 dB lower if the boundary absorption coefficient is increased from 0.1 to 0.9. |
| Kang, 2002 [51] | n/a | SI | SPL, RT30 | n/a | • Sound attenuation along the length of the canyon is significant, typically at 20–30 dB/100 m.<br>• Over 2–4 dB extra attenuation can be obtained either by increasing boundary absorption evenly or by adding absorbent patches on the façades or the ground. Reducing building height has a similar effect.<br>• A gap between buildings can provide about 2–3 dB extra sound attenuation, especially in the vicinity of the gap.<br>• The effectiveness of air absorption on increasing sound attenuation along the length could be 3–9 dB at high frequencies. |
| Iu and Li, 2002 [52] | n/a | SI, FM, SM (1:10) | EA, TL | n/a | • Sound propagation in cities involves phenomena, such as reflections and scattering at the building façades, diffusion effects due to recessions and protrusions of building surfaces, geometric spreading, and atmospheric absorption. |
| Picaut and Simon, 2001 [53] | Nantes, France | FM, SM (1:50) | RT, sound attenuation | n/a | • Façade geometry affects reverberation and sound propagation. The architectural (geometrical) complexity of building façades is the fundamental cause of sound diffusion in streets. |
| Chew and Lim, 1994 [54] | Singapore | SI, FM | L10 | n/a | • Buildings on one side of the expressway increase the L10 by 2.5 dB(A) at 1 m from the façade, while buildings on both sides could increase the L10 by more than 10 dB(A). The façade effect is significant only when one is near the buildings. At distances of more than 20 m from the buildings, the façade effect is negligible.<br>• The so-called cannon effect, in which L10 increases with the height of the buildings, is significant only when the buildings are close together, say, less than 20 m apart. The diffuse energy component dominates, increasing the overall L10 by 7–11 dB(A). |

**Table 2.** Summary of façade parameters and contextual parameters described in the reviewed studies (n = 40) in reverse chronological order of publication. Abbreviations: not-applicable (n/a); reflective material (RM); absorptive material (AM); vertical greenery system (VGS); background noise (BN); instrument (I); people (P); simulation (SI); number of participants (n).

| Reference | Façade Parameters | | | Contextual Parameters | | | | | | | |
|---|---|---|---|---|---|---|---|---|---|---|---|
| | Geometry | | Materials | Sound | | | Path | | | Receiver | |
| | Height | Depth, Inclinations | | Source | Frequency | BN | Boundary Conditions | Materials | Atmospheric Conditions | Position | Type |
| Masullo et al., 2021 [15] | 4 floors | Flat with balconies | RM (Concrete, glass, stone) | AC split units on façade | up tp 10 kHz | Yes | Historic city centers, around 4 floors | RM | n/a | First floor balcony | P (n = 30) |
| Niesten et al., 2021 [16] | 8, 18 floors (50 m) | Flat, angled upwards, balconies | RM, AM | Road traffic | 12.5–31 kHz | Yes | Courtyard between buildings near a busy road, trees | RM, AM (brick, glass, asphalt, grass, water) | 20 °C, 50% | Around the building, height 4 m | I, SI |
| Montes González et al., 2020 [17] | 8 m | Flat | RM | Road traffic | Broadband | Yes | Street with two traffic lanes and parked cars | RM | n/a | Vertically along the façade | I, SI |

**Table 2.** *Cont.*

| Reference | Façade Parameters | | | Contextual Parameters | | | | | | | |
| | Geometry | | | Sound | | | Path | | | Receiver | |
| | Height | Depth, Inclinations | Materials | Source | Frequency | BN | Boundary Conditions | Materials | Atmospheric Conditions | Position | Type |
|---|---|---|---|---|---|---|---|---|---|---|---|
| Cabrera et al., 2020 [18] | 24 m | Balconies, recessed windows | RM | Road traffic | 315–12,500 Hz; Simulation: up to 25,900 Hz | Yes | Mid-rise, high-rise | RM, AM | (a) 14 °C, 65%, (b) 21 °C, 59%, (c) 24 °C, 84% | By the sound source | SI, I |
| Hupeng et al., 2019 [19] | 20 m | Flat | RM (Brick with plaster) | Road traffic | 1 kHz | n/a | Canyons in high-density city | RM (asphalt, brick walls with plaster) | n/a | Both sidewalks, height 1.50 m | SI |
| Leistner et al., 2019 [20] | n/a | Flat | RM, AM, VGS | Road traffic | n/a | n/a | Buildings next to motorway | RM | n/a | Mapped around the buildings | SI |
| Yu et al., 2019 [21] | Mid-rise | Flat | RM (rough and fluctuated) | Road traffic | 500 Hz and 1000 Hz | Yes | Canyons | RM (glass, concrete) | 3–12 °C, 72–85% | Central street axis, height 1.25 m | SI, I |
| Taghipour et al., 2019 [22] | up to 7 floors | Flat and balconies | RM, AM | People talking, basketball, children | Broadband | Yes | Courtyard housing complex | RM, AM (brick, glass, concrete, grass) | n/a | Courtyard | P (n = 27) |
| Badino et al., 2019 [23] | 17.8 m | Loggias and balconies | RM, AM | People talking | up to 8000 Hz | n/a | Canyon | RM | n/a | In balconies on both sides of the street height 1.5 m | SI |
| Calleri et al., 2018 [24] | 2–4 floors | Flat and diffusive in multiple orientations | RM, AM, VGS (Plastered brickwork, concrete, green wall) | People talking | 63–8000 Hz | Yes | Public octagonal square (2000 m²) | RM | n/a | On public square | P (n = 31) |
| Jones and Goehring, 2018 [25] | 33 floors | Flat | RM (Perforated metal panels) | Wind | Broadband | Yes | High-rise | RM | Wind (southwest and south, >10 m/s) | On the façade | SI |
| Montes González et al., 2018 [26] | 3 floors | Flat | RM | Road traffic | Broadband | Yes | Single building next to a street with parked cars | RM | n/a | Vertically along the façade, heights 1.5, 4 and 7.3 m | I, SI |
| Qu and Kang, 2017 [27] | 8 m | Flat | RM | Wind turbines | Broadband | n/a | Medium-density neighborhoods | RM | 10 °C, 70% | 0.5 m from façades | SI |
| Flores et al., 2017 [28] | Mostly 2 floors | Flat | RM | Air traffic | n/a | Yes | Mostly two-story buildings near airports | RM, AM | (a) 16.4 °C, 1.9 m/s; (b) 12 °C, 2.7 m/s. | Around buildings and in free field | I |
| Echeverria Sanchez et al., 2016 [29] | 8 floors (25.6 m) | Flat, downwardly inclined, upwardly inclined, convex, concave. | RM, AM (Glass, brick) | Road traffic | Broadband | n/a | Canyons, sound barriers | RM | n/a | Sidewalk | SI, I |

**Table 2.** *Cont.*

| Reference | Façade Parameters | | | Contextual Parameters | | | | | | | |
| | Geometry | | | Sound | | | Path | | | Receiver | |
| | Height | Depth, Inclinations | Materials | Source | Frequency | BN | Boundary Conditions | Materials | Atmospheric Conditions | Position | Type |
|---|---|---|---|---|---|---|---|---|---|---|---|
| Jang et al., 2015 [30] | 10 m | Flat | RM, AM, VGS (Glass, brick, green wall) | Road traffic | 1 kHz–40 kHz (scaled 1:10) | n/a | Canyon, two-lane road surrounded by three-floor buildings | RM, AM, VGS | n/a | Parallel to the façade at 1.6 m, height 1.5, 4.5, 7.7, 10.9 m | I, SI |
| Can et al., 2015 [31] | 10–30 m | Flat, diffusive | RM | Road traffic | n/a | n/a | Canyon | RM | Atmospheric absorption | Center of the street | SI |
| Guillaume et al., 2015 [32] | 17, 15 m | Flat | RM, AM, VGS (Concrete, glass, green wall) | Road traffic | up tp 1000 Hz | n/a | Canyon of infinite length | RM | Atmospheric absorption | Along the façade | SI |
| Sakamoto and Aoki, 2015 [33] | 5 floors, 20 floors | Flat with horizontal eaves/louvers | RM (Aluminum eaves/louvers) | Road traffic | up to 2000 Hz | n/a | Mid-rise urban area | RM (MDF board in scale model) | n/a | On the façade between eaves | I, SI |
| Jang et al., 2015 [34] | 3 floors | Flat | RM, AM, VGS (brick, vegetation) | Road traffic | up to 4000 Hz | n/a | Canyon and courtyard | RM, AM (asphalt, brick, heavy grass) | Atmospheric absorption | Sidewalk and courtyard, height 1.5 | I |
| Hao and Kang, 2014 [35] | n/a | Flat | RM, AM (masonry, glass) | Air traffic | 630, 1600 and 3150 Hz | n/a | low-density residential areas | RM | n/a | Mapped around buildings, height 1.6 m | SI |
| Silva et al., 2014 [36] | 4 floors (12 m) | Flat | RM | Road traffic | n/a | n/a | Residential areas | RM, AM | 15 °C, 70% | Grid along the façade | SI |
| Van Renterghem et al., 2013 [37] | 19.2 m | Flat | RM, AM, VGS (Brick, glass, green wall) | Road traffic | Broadband | n/a | Canyon and courtyard | RM | Atmospheric absorption | In the street and along the façades | SI |
| Thomas et al., 2013 [38] | 5–20 m | Flat | RM | Road traffic | 63 Hz–16 kHz | Yes | Canyons (99 streets) | RM | 5.0 °C, 80% | By the sound source | I, SI |
| Hornikx and Forssén, 2011 [39] | 18 m | Flat | RM, AM | Road traffic | up to 500 Hz | n/a | Canyons, courtyard (with gaps) | RM, AM | n/a | Grid in the street and vertically along the façade, height 1.5 m | SI |
| Tang and Piippo, 2011 [40] | 8 m | Flat and inclined at 60°, 70°, 80° | RM | Road traffic | up to 5000 Hz | n/a | Canyon | RM (wood in scale model) | n/a | Along the façade | I |
| Oliveira and Silva, 2011 [41] | 4–8 floors | Flat | RM | Road traffic | n/a | n/a | Buildings between two parallel streets | RM | n/a | Mapped around the buildings | SI |
| Okada et al., 2010 [42] | 30 m | Flat | RM | Road traffic | 250 Hz–1 kHz | n/a | Canyon with viaduct | RM | n/a | Sidewalk, height 1.5 m and 4.5 m | SI, I |
| Hornikx and Forssén, 2009 [43] | 10 m, 20 m | Flat | RM, AM (plaster, glass, brick, wood, plaster) | Road traffic | 100 Hz–1000 Hz | n/a | Canyons | RM | Yes | In the street and along the façades | SI |

**Table 2.** *Cont.*

| Reference | Façade Parameters | | | Contextual Parameters | | | | | | | |
| | Geometry | | Materials | Sound | | | Path | | | Receiver | |
| | Height | Depth, Inclinations | | Source | Frequency | BN | Boundary Conditions | Materials | Atmospheric Conditions | Position | Type |
| Hornikx and Forssén, 2008 [44] | 20 m | Flat | RM, AM (absorptive and diffusive) | Road traffic | 100 Hz–1000 Hz | n/a | Canyons, courtyard | RM | 20 °C | Sidewalk | I |
| Onaga and Rindel, 2007 [45] | 6, 18, 30 m | Flat | RM, AM | Road traffic | 250 Hz, 2500 Hz | n/a | Canyon | RM | n/a | Sidewalk | SI |
| Heimann, 2007 [46] | 6 m | Flat | RM, AM | Road traffic | 250 Hz | No | Single street and parallel streets | RM, AM | Wind (8 m/s at height 10 m) | Mapped around the building | SI |
| Lee et al., 2007 [47] | 15 floors | Flat and balconies | RM, AM | Road traffic | 500 Hz–1 kHz | Yes | Apartment complex located near a six-lane road. | RM, AM | Atmospheric absorption | In the balconies | SI, I |
| Van Renterghem et al., 2006 [48] | 10 m | Flat, with protrusions, balconies, inclined parapets | RM | Road traffic | up to 1250 Hz | n/a | Canyon | RM | Wind | Mapped around the building | SI |
| Thorsson and Ogren, 2005 [49] | up to 6 floors | Flat | RM, AM | Road traffic | n/a | n/a | Parallel canyons | RM | n/a | On the façade | SI |
| Kang, 2005 [50] | 6–50 m | Flat | RM (diffusive and reflective) | Road traffic | n/a | n/a | Urban squares | RM | Atmospheric absorption | In the square, height 1.2 m | SI |
| Kang, 2002 [51] | 8 m | Flat | RM (diffusive and reflective) | Road traffic | 400 Hz–16 kHz | n/a | Canyon | RM | Atmospheric absorption (20 °C, 40–50%) | In the street | SI |
| Iu and Li, 2002 [52] | 18 m | Flat | RM | Road traffic | up to 6000 Hz | n/a | Canyon | RM (wood in scale model) | Atmospheric absorption | Sidewalk | SI, I |
| Picaut and Simon, 2001 [53] | 8 m, 12 m | Flat | RM (wood in scale model) | Road traffic | 250–5000 Hz | n/a | Canyon (96 m length, 8 m height and 12 m wide) | RM (wood in scale model) | Atmospheric absorption | Along the street | I |
| Chew and Lim, 1994 [54] | 15 floors (45 m) | Flat | RM | Road traffic | 500 Hz | n/a | Canyon | RM | n/a | up to 200 m from façade. Height: 1–5 m | SI |

*3.1. Study Design*

The general methods used to study façade-related effects on the acoustic environment are here classified into four groups: field measurements, scale model measurements, simulation models, and laboratory experiments. Over half of the studies used simulation models, at times carried out in parallel with field measurements or scale model measurements to validate the methods. Laboratory experiments that included human participants were conducted in only three studies [15,22,24].

As seen in Table 1, all of the 40 studies used at least one decibel-based measurement, including sound pressure level (SPL), equivalent continuous sound level (Leq), average sound level (Lavg), day–evening–night equivalent sound level (Lden), sound level exceeded for 10% of the time of the measurement period (L10), insertion loss, sound attenuation, reverberation time (RT, or T), and early decay time (EDT), among others. Only three studies [15,22,24] used perceptual attributes, such as perceived loudness, noise annoyance, acoustic comfort, and perception of space wideness. These studies were carried out with real participants in Italy, Spain, and Switzerland with groups of 27–31 people surveyed per study.

*3.2. Reported Effects*

In Table 1 the effects reported by each author related to façades, the acoustic environment, and the soundscape are presented. In Table 3, the effects are classified into three main groups and seven sub-groups, and are discussed in detail in the next section.

**Table 3.** Classification of reported effects related to façades, the acoustic environment and/or the soundscape identified in the literature review.

| Effects of façades on the urban acoustic environment | References |
|---|---|
| Sound-reflecting effects | |
| • Sound pressure level affected by façade height, depth, and inclination | [16,18,19,23,29,33,38,40–42,47,50,53,54] |
| • Reverberation time affected by façade height, depth, and inclinations | [19,38,40,47,50,53] |
| • Sound pressure level affected by façade diffusion/scattering | [21,31,42,44,45,48,50,53,54] |
| • Reverberation time affected by façade scattering | [21,45] |
| Sound-absorbing effects | |
| • Sound pressure level affected by façade absorption | [16,20,22–24,30,32,34,35,37,39,43–45,47–50] |
| • Reverberation time affected by façade absorption | [24,34,44,45,47,51] |
| Sound-producing effects | |
| • Sound produced at the façade by mechanical equipment | [15] |
| • Sound produced at the façade by wind-induced vibration | [25] |
| **Effects of façades on the urban soundscape** | **References** |
| Auditory effects | |
| • Perception of space wideness affected by façade absorption | [24] |
| • Perceived acoustic comfort affected by façade absorption | [22] |
| Non-auditory effects | |
| • Perception of loudness affected by façade visual pleasantness | [15] |
| • Noise annoyance affected by façade visual pleasantness | [15] |
| **Effects of the context on the acoustic environment around façades** | **References** |
| Boundary effects | |
| • Sound pressure level around façade affected by urban morphology | [27–29,35,36,39,40,42,44,46,48,51,54] |
| • Sound pressure level around façade affected by shielding | [17,26,29,37,39,44,49] |
| Atmospheric effects | |
| • Sound pressure level around façade affected by wind | [25,46,48] |

*3.3. Façade Parameters*

For the purpose of this study, "façade geometry" considers the three-dimensional characteristics of height, depth, and inclination of the façade. Studies that considered "flat" geometries (referring to a single vertical plane oriented at a 90° angle to the ground with no relevant depth) are the most common in the review; however, several studies

considered balconies, inclined walls or parapets, and façades with recessed windows or other architectural elements. Geometrical changes in terms of depth and upward inclination show capabilities of sound pressure level reduction by up to 9.3 dB(A) [16] compared to a "flat" façade. The height of façades showed to influence the acoustic environment significantly, as multiple studies concluded that higher façades led to higher sound pressure levels [38,41,42,50,54] and higher reverberation time [19,50], mainly due to sound reflection, diffusion, and scattering on façades. The lowest façades reported among the studies had heights of 5–6 m in buildings with two floors [24,28,38,45,46,50]. The highest façade surveyed by field measurements was a building with eight floors with a height of 24 m [18], and the highest façade simulated by computational methods [25] was a building with 33 floors approximately 100 m high.

The materials identified in the studies were classified as "reflective", including concrete, glass, masonry, stone, brick, plaster, and wood, among others, and as "absorptive", including vertical greenery systems, perforated metal panels, and porous sound-absorbing tiles. In some studies, materials were not specified but were described as reflective or absorptive. Façade absorption was among the most studied topics, found in 18 of 40 articles related to façade materials affecting sound levels and reverberation time on the street.

The acoustic performance of vertical greenery systems (VGS) proved to be efficient for reducing road traffic noise in streets, offering sound pressure level reductions by up to 5 dB(A) and reductions in reverberation time. Additionally, façade vegetation can produce auditory effects on the soundscape by altering the perception of acoustic comfort and space wideness through sound absorption [22,24].

### 3.4. Contextual Parameters

The previous section focused on the characteristics of façades, but the effects of façades are also dependent on the context where they are located. Information regarding the individual context of each study was organized into three aspects: sound, path, and receiver.

#### 3.4.1. Sound

The most recurrent sound source among the selected papers was road traffic, considered as the main source in 32 studies. People's voices were considered as sound sources three times [22–24], air traffic was considered twice [28,35], and sound generated by wind [25], wind turbines [27], or air conditioning (AC) units [15] on façades were studied once, respectively. The movement of sound sources, such as road traffic or airplanes. is studied in simulations and scale models as one or a series of static omnidirectional sources. Background noise was considered in 12 articles, and is often not included in the design of most experiments, especially when performing simulations and scale model measurements rather than field measurements.

#### 3.4.2. Path

The propagation path between sound sources, façades, and receivers considers the physical boundaries created by the terrain and the infrastructure, as well as the atmospheric conditions on the site. In most of the studies, the geometrical complexity of buildings is reduced to simple surfaces with no significant depth, altering the way that sound is modified between source and receiver. The most recurrent urban profile identified in the studies is street canyons, followed by courtyards, and then residential neighborhoods, historic city centers, and urban squares. The materials in urban environments were most of the time composed of hard surfaces with low absorption coefficients and one study considered water as a reflective surface. Only 10 studies included absorptive urban surfaces, such as grass. Other infrastructure that affected sound levels at the location of the façade included a viaduct vertically parallel to the street [42], and parked vehicles between the sound source and the façade [17,26]. Additionally, wind effects were studied by [25,46,48].

### 3.4.3. Receiver

Three types of receivers were defined: instruments that measure or capture the acoustic environment, for example, a sound level meter or a microphone; people participating in a laboratory experiment by filling a survey; and sound at the receiver that is simulated by numerical or analytical methods. The most recurrent types of receivers were simulated and measuring instruments. Only three studies explicitly considered the soundscape approach, involving real people as receivers [15,22,24].

## 4. Discussion

The information provided by the selected literature regarding the effects of façades was classified into three main groups and seven sub-groups of effects as seen in Table 3. This classification is proposed exclusively with data from the 40 articles included in the systematic review and it was sufficient to identify similarities, contrasts, and patterns between the studies, and organize the findings in a structured manner.

### 4.1. Effects of Façades on the Urban Acoustic Environment

These effects were identified in 33 of 40 articles that studied these façade effects by employing decibel metrics via field measurements, scale model measurements, or simulations.

#### 4.1.1. Sound-Reflecting Effects

The concepts of the "façade effect" and "cannon effect" are mentioned by the article date the furthest back in this systematic review (1994) [54] and they refer to the increase in sound pressure level at a location due to the multiple reflections from the ground and vertical surfaces on urban environments. It is pointed out that the façade effect is more significant when the receiver is near the buildings, and that the effect is negligible at distances of more than 20 m from the buildings. In their simulation model, the presence of a building on one side of the street increased the $L_{10}$ by 2.5 dB(A) at 1 m from the façade, while parallel flat buildings on both sides increased the $L_{10}$ by more than 10 dB(A). In the scale model studied by [40], two opposite walls raised sound levels by 8 dB, and inclining one of the surfaces in the canyon by up to 60° upwards decreased the reverberation strength making the sound field less uniform. Strategies modifying façade geometries to reflect sound upwards presented by [16] reached a reduction in sound pressure level by up to 9.3 dB(A). When the height of buildings increases, generally the sound pressure level also increases, [39,41,42] as well as the mean reverberation time [19,50]. Street reverberation can strongly increase sound pressure levels in urban streets [37].

The depth of balconies studied by [23] presented a reduction up to 2.8 dB for receivers in the balcony when increasing the depth from 0.9 to 1.5 m. The effects of flat eaves/louvers in horizontal direction were studied by [33], concluding that flat eaves do not reduce noise significantly unless their lower surfaces are inclined outwards.

One study conducting field measurements and simulations [18] focused on façade retroreflections, which occur when sound is reflected back to the source, concluding that the increase in sound pressure levels next to the source due to retroreflections is most prominent in the high-frequency range. Additionally, the effect of retroreflections is measurable but not necessarily audible in the presence of high background noise.

Façade diffusion effects related to the spreading of sound energy on the environment are more prominent when buildings are close together [54] and are caused mainly by the geometry (architectural complexity) of façades, leading to an increase in sound pressure levels in streets, the SPL in a simulation by [31] raised by up to 10 dB due to diffusion.

#### 4.1.2. Sound-Absorbing Effects

While some of the most common construction materials in cities can be considered sound-reflective and have generally low absorption properties (concrete, glass, brick, metal, and asphalt), sound-absorptive materials on façades with higher absorption coefficients were studied in multiple urban scenarios. Façade materials with higher absorption coeffi-

cients present a potential for reducing sound pressure levels and reverberation without changing the building geometry significantly. Covering façades with absorbing materials resulted in noise level reductions of 4–5 dB inside street canyons [49,51]. Sound absorbing design strategies studied by scale models and simulations by [16] reduced SPL from 1.7 to 5.5 dB(A), and combining façade absorption with geometrical changes reduced the SPL by up to 9 dB(A). Noise reduction by up to 10 dB measured at the façade due to an optimized geometry cladded with sound-absorbing materials was reached in a simulation by [23].

This review paper classifies vegetation on the façades as an absorptive material; however, the geometrical properties of vegetation on façades could be relevant and require attention. Vertical greenery systems were considered in six studies [20,24,30,32,34,37] and offered noise reductions between 2 and 5 dB(A). Applications of vegetation on the upper halves of buildings surrounding the street have shown reductions of about 4 dB [37,43] and the use of vegetation in the lower half of the buildings only resulted in a limited reduction. In the case of a street canyon next to a courtyard [16], fully greening the courtyard façades was beneficial compared to only putting a green wall in the upper half.

### 4.1.3. Sound-Producing Effects

Façades can also affect the acoustic environment by acting as sound sources as seen in two cases [15,25]. Wind-induced noise and vibrations can occur due to the urban morphology, façade orientation towards wind direction, façade geometry, and materials. The corners and top of the building are the most critical areas when considering wind-induced noise on façades [25]. Air conditioning (AC) units are common façade elements that generate noise and were included in the perceptual study of [15].

### 4.2. Effects of Façades on the Urban Soundscape

Only three studies from the systematic review [15,22,24] applied the soundscape approach focusing on people's perception of the acoustic environment, instead of using the decibel unit only. The empirical research conducted in these three papers included laboratory experiments involving real people being surveyed to describe the acoustic environment. They included 27–31 participants per study, and the locations were Italy, Spain, and Switzerland. People's characteristics beyond nationality are not included in the scope of this review due to the low number of studies in the field. However, people's demographic characteristics could influence the outcome of perceptual experiments. Social conditions could be implemented as a contextual parameter, along with boundary and atmospheric conditions when studying façade effects on the soundscape.

It was noticed that there were no studies using psychoacoustic parameters (e.g., loudness, roughness, sharpness, others), which are frequently employed in soundscape research, addressed in Part 2 of the soundscape standard [10].

### 4.2.1. Auditory Effects

Listening tests were conducted by [22] in order to study the influence of façade absorption on people's short-term acoustic comfort and discomfort in a courtyard in Switzerland. The results of surveying 27 participants suggest that the use of what the authors called "moderate absorption" of the building façade increased people's self-reported acoustic comfort in a courtyard, in comparison to reflective surfaces. "Exaggerated absorption" (covering every surface with absorptive material) was not found to be significantly more effective for acoustic comfort than a façade with "moderate absorption".

The influence of façade absorption on the perception of space wideness was studied by [24], surveying 31 participants in Italy with an online test in which each page presented an audio track that participants could play as many times as needed, and fill-in a 5 point scale to provide the answer. The pages were randomized in order to avoid order effects. Participants could move back to the previous pages in order to listen again to each auralization, if they needed. It was observed that the absorption and scattering coefficients of

façades influence the acoustic characteristics at the receiver's position by being correlated to the reverberation time and SPL of the space.

### 4.2.2. Non-Auditory Effects

The visual impact of air conditioning units on the façades of European historic city centers was studied using virtual reality (VR) to survey people's perception of the acoustic environment [15], with a sample of 30 participants in Italy and Spain. In the audio-visual experiment, air conditioning units that were installed in the façades without any aesthetic considerations led to a higher perception of loudness and noise annoyance compared to situations with more aesthetically integrated elements, suggesting that there is a strong interconnection between visual and auditory stimuli.

### 4.3. Effects of the Context on the Acoustic Environment around Façades

While individual façade properties influence their surrounding acoustic environment (as seen in Section 4.1), contextual parameters (in this study considered as sound, path and receiver) influence the acoustic environment around the façade. The sound sources in this study were considered contextual parameters, however, the conceptual framework of [5] classifies sound sources as an individual element independent from the context.

### 4.3.1. Boundary Effects

Urban morphology, including the buildings in the area, the streets, sidewalks, and other infrastructure, can lead to significant variations in noise levels around the façade. A simulation study [27] found a decrease of up to 10 dB(A) around dwellings with different morphologies in the distant range of 400–1000 m from the sound source. Gaps between buildings along a street provided about 2–3 dB of sound attenuation, especially in the vicinity of the gap [51]. An opening in a courtyard led to a noise increase inside the courtyard of up to 7.2–10 dB(A) compared to a closed courtyard [39].

### 4.3.2. Atmospheric Effects

The atmospheric conditions mentioned in the studies included temperature, humidity, and wind; however, in half of the studies these are not specified. The effects of atmospheric absorption on sound propagated along a street length could be 3–9 dB at high frequencies [51]. Additionally, it is noticed that boundary conditions also led to atmospheric effects, as it was observed that urban morphology and street geometry can greatly influence wind flow [27]. As described by [46] the wind effect in a scenario of parallel streets is more pronounced for hip-roof buildings. In the case of parallel streets, upwind facades are slightly louder than downwind facades because they are simultaneously exposed to downwind propagating sound from the next parallel street.

## 5. Conclusions

This paper presented a systematic literature review of the effects of façades on the urban acoustic environment and the soundscape, as well as the effects of the context on the acoustic environment around façades. Data from 40 peer-reviewed articles were extracted to answer the research questions related to (1) the study design used in published literature, (2) the parameters of façades and the context considered in the experiments and (3) the effects of façades on urban acoustic environment and soundscape.

Firstly, the effects of façades on urban acoustic environments have been increasingly studied for over two decades by multiple research methods (in this study organized as field measurements, scale model measurements, laboratory experiments and simulations), providing a general understanding of façade influence on sound propagation outdoors, and how the context influences the acoustic environment. On the other hand, the effects of façades on the soundscape have only been published in the past five years and only three studies in this review conducted experiments involving real people (groups of 27–31 participants) as descriptors of the acoustic environment.

Secondly, the parameters of façades that can be associated with the production of effects include façade geometry and façade materials. The effects of the context on the acoustic environment around façades include boundary conditions and atmospheric conditions. Social (sociodemographic) conditions could have an influence on the effects of façades on the soundscape, however, more research on façades using peoples' opinions as descriptors of the acoustic environment are necessary in order to have comparable data. It can be concluded that the effects of façades on the urban acoustic environment and the soundscape are not absolute (or applicable to every façade), but relative to the context in which the façades are located.

Thirdly, the effects reported in 40 articles were classified into three groups: effects of façades on the urban acoustic environment (sound-reflecting, sound-absorbing and sound-producing effects); effects of façades on the urban soundscape (auditory and non-auditory effects); and effects of the context on the acoustic environment around façades (boundary effects and atmospheric effects). In the reviewed studies, sound-reflecting effects reached a reduction of up to 10 dB(A), as well as reduced reverberation time. Sound-absorbing effects led to reductions of up to 5.9 dB(A) and reduced reverberation time. Sound-producing effects in this review were caused by vibrations on façade panels due to wind, as well as noise produced by air conditioning units. Auditory effects were reported by participants who's perception of space wideness and acoustic comfort was affected by façade absorption in listening experiments. Non-auditory effects were reported by people participating in an audio-visual experiment in which participants' perception of loudness and noise annoyance was influenced by visual stimuli. The boundary effects and atmospheric effects in urban environments have a significant influence on the acoustic environment and can account for up to 2–10 dB(A) around the façade.

## 6. Limitations

The extraction of data in this review was limited to three databases (Scopus, Science Direct, Web of Science) surveyed in October 2021 using a predefined combination of search terms, as mentioned in Section 2. The authors acknowledge the possibility of other studies related to the topic not being included in this review by not being accessible through the databases employed, not being identified by the designated search terms, or not passing the eligibility criteria.

**Supplementary Materials:** The following supporting information can be downloaded at: https://www.mdpi.com/article/10.3390/su14159670/s1, File S1: PRISMA 2020 Checklist [55].

**Author Contributions:** Conceptualization, A.B., A.P., J.K. and A.L.-N.; data collection, screening and reviewing, A.B.; writing of the manuscript, A.B.; supervision, U.K. and D.A. All authors have read and agreed to the published version of the manuscript.

**Funding:** This research received no external funding.

**Institutional Review Board Statement:** Not applicable.

**Informed Consent Statement:** Not applicable.

**Conflicts of Interest:** The authors declare no conflict of interest.

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
