# Peer review of "Effects of Façades on Urban Acoustic Environment and Soundscape: A Systematic Review"

_sustainability, doi:10.3390/su14159670_

Round 1

Reviewer 1 Report

The article presents a systematic literature review of the effects of façades on the urban acoustic environment and soundscape and contextual parameters on the acoustic environment around façades. Analyzed data from 40 articles, identifying the assessment methods used, facade parameters, contextual parameters, and reported effects. The goal was clearly formulated, which was supported by the findings of the conducted literature analyses.

The article is a great study and will undoubtedly be used in manuscripts on similar topics. But why are two of these selected articles only from 2021? Three of 2020. Are there no more valuable studies on the subject of the subject?

To sum up - the subject of the article is clearly presented, the goal has been defined, and the conclusion was formulated correctly. Therefore, minor linguistic errors do not affect the readability of the article.

Congratulations to the Authors.

Author Response

We appreciate your feedback and positive evaluation. Please see the attachment.

Reviewer 2 Report

Major commend:

1. In lane 94, please re-clarify the term “Noise” as a keyword for searching the article which is related to the sound source, it is explained and considered to the resource, not an unfriendly factor, suggest to discuss it more.

2. In Figure 2 which presents the number of publications published from 1994 to 2021, differentiating between studies that focus on noise more than studies that focus on soundscape? Can you explain that trend more?

3. Subjective studies with laboratory experiments including human participants were conducted in only three studies, what if using the terms “hearing perception or acoustical comfort” for searching will gather more results?

4. The concept of the “façade effect” is affected by the percentage of surrounding the geometrical enclosure mentioned by the article that dates the furthest in this systematic review (1994) [54] . The sound source is referring to a linear or moveable (Expressway) rather than a point source or fixed, any studies discuss the differences of character between two sound sources?

5. In addition to the above sound source characteristics, geometric street space, and vertical facades with boundary closure which are included plane enclosure and vertical enclosure also be considered.

Minor revision:

In Lane 83, Section 4 is a discussion of the “classified” of effects. “Classified” should be corrected as “classification”

In Lane 103, ( façade acoustics, soundscape research) should be added the “and” ( façade acoustics and soundscape research).

In Lane 126, abbreviation of background noise might be as (BN)

In Table 1, Acoustic retrorefection should be corrected as retroreflection.   

In Table 2, “MDF” in scale model might be “MDF board”.   

In Lane 137, objective measures used, perceptual attributes considered (if any). should be added the “and” ( objective measures used and perceptual attributes…).

In Lane 169, reduction “in”… should be erased.

In lane178 the studies “was” classified as… should be corrected as “were”.

In lane 183, found 18 in “out” of 40 … should be erased.

Author Response

The authors appreciate the feedback provided by the reviewer. Please see the attachment.

Reviewer 3 Report

This review paper presents the state-of-the-art about the acoustic effects of building facades within urban contexts, which is quite timely in this developing area. Most of the relevant issues are considered in the rather compact review vis-à-vis the extant research. I have but minor comments on this review paper:

- The discussion section seems a bit rushed, especially section 4.1, and had more errors than the rest of the paper (covered in the following), which was otherwise relatively free of errors. The authors can consider expanding the text in this section (especially 4.1) a bit by perhaps including some more contextual details about the papers included. However, this is a minor suggestion.

- Linked to the above, the conclusion section seems a bit too long and not structured very well from L347 onward. Perhaps some of these points can go in the discussion section to have crisper conclusion points instead of long text as is currently the case.

- L84: ‘classification’

- Table 1, ref [33], ‘…eave reflects and incident…’: Either there is a word missing or the 'and' should be 'the'  

- Table 1, ref [46]: ‘hip-roof’ in the first line.

- Table 1, ref [49]: ‘by’ instead of ‘with’?

- L151: ‘at’ instead of ‘in’?

- L169: delete ‘in’?

- L170: ‘Façade height was shown’?

- L178: ‘were’ instead of ‘was’?

- L233: ‘a’ instead of ‘the’?

- L258: There seems be some missing text after ‘absorption coefficient is’  

- L264-265: Change this sentence.

- L366-368: Although this seems like an obvious statement to make, it is abruptly presented in the conclusion without being covered elsewhere. I would strongly recommend presenting at least some research about this aspect (if present, if not then simply acknowledging this would do) in a previous section, followed by a short sentence in conclusions.

- L373: Limitations should ideally be a part of the manuscript text, most likely in the discussion, unless there is a journal rule.

Author Response

We appreciate the feedback provided. Please see the attachment.

Round 2

Reviewer 2 Report

Accept by the editor's final decision.